# Network Analysis of Anxiety Symptoms in Front-Line Medical Staff during the COVID-19 Pandemic

**DOI:** 10.3390/brainsci13081155

**Published:** 2023-08-01

**Authors:** Lin Wu, Lei Ren, Fengzhan Li, Kang Shi, Peng Fang, Xiuchao Wang, Tingwei Feng, Shengjun Wu, Xufeng Liu

**Affiliations:** 1Department of Military Medical Psychology, Air Force Medical University, Xi’an 710032, China; wlfmmuedu@163.com (L.W.);; 2Military Psychology Section, Logistics University of PAP, Tianjin 300309, China; 3Military Mental Health Services & Research Center, Tianjin 300309, China

**Keywords:** COVID-19, front-line medical staff, GAD-7, anxiety, network analysis

## Abstract

Background: This research analyses the relations between anxiety symptoms from the network perspective to deepen the understanding of anxiety in front-line medical staff during the COVID-19 pandemic and can also provide a reference for determining potential goals of clinical interventions. Methods: A convenience sampling was adopted, and the Generalized Anxiety Disorder 7-item scale (GAD-7) was administered to front-line medical staff through online platforms. A regularized partial correlation network of anxiety was constructed and then we evaluated its accuracy and stability. The expected influence and predictability were used to describe the relative importance and the controllability, using community detection to explore community structure. The gender-based differences and the directed acyclic graph were implemented. Results: The connections between A1 “Feeling nervous, anxious or on edge” and A2 “Not being able to stop or control worrying”, A6 “Becoming easily annoyed or irritable” and A7 “Feeling afraid as if something awful might happen”, etc., were relatively strong; A2 “Not being able to stop or control worrying” and A3 “Worrying too much about different things” had the highest expected influence, and A2 “Not being able to stop or control worrying” had the highest predictability. The community detection identified two communities. The results of the gender network comparison showed the overall intensity of the anxiety network in women was higher than that in men; DAG indicated that A2 “Not being able to stop or control worrying” had the highest probabilistic priority; the lines from A2 “Not being able to stop or control worrying” to A1 “Feeling nervous, anxious or on edge” and A2 “Not being able to stop or control worrying” to A7 “Feeling afraid as if something awful might happen” represented the most important arrows. Conclusion: There exist broad interconnections among anxiety symptoms of front-line medical staff on the GAD-7. A2 “Not being able to stop or control worrying” might be the core symptom and a potential effective intervention target. It was possible to bring an optimal result for the entire GAD symptom network by interfering with A2 “Not being able to stop or control worrying”. GAD may have two “subsystems”. The modes of interconnection among anxiety may be consistent between genders.

## 1. Introduction

Coronavirus disease 2019 (COVID-19) is a serious infectious disease caused by a novel coronavirus and is characterized by a sudden onset, a high infectiousness, long latent periods, and a strong pathogenicity. As of 6 May 2022, the cumulative cases across the globe amounted to 515,568,625, and the number of deaths was 6,276,443. The COVID-19 pandemic has caused serious human harm and was declared a Public Health Emergency of International Concern by the World Health Organization (WHO) [1,2].

During the COVID-19 pandemic, medical staff have been the core force in treating patients and therefore bear substantial responsibility and pressure. In this situation, their mental health will inevitably be affected [3,4,5]. Emotion, as an important part of the mental health of medical staff, has been confirmed to be the serious aspect affected by the COVID-19 pandemic. This study focuses on the emotional issues of front-line medical staff during the COVID-19 pandemic.

The impact of the COVID-19 pandemic on medical staff’s emotions is mainly reflected in anxiety, fear, and depression. Liu et al. reports that 44.7% of medical staff exhibit anxiety and 36.1% show fear [6]. A study by Liu et al. found that up to 50.58% of medical staff exhibited anxiety [7]. Xu and Zhang found that 51.22% of front-line medical staff experienced the feeling of fear and that 39.02% exhibited anxiety [8]. In a study by Wu et al., the proportion of medical staff who exhibited anxiety and depression symptoms was 18.1% and 19.2%, respectively [9]. A study conducted by Liu et al. in mid-February of 2020 found that of the 512 medical staff who participated in the study, 12.5% showed anxiety symptoms [6]; this percentage was lower than that reported by Lai et al. for the same time period (44.6%) [10]. Lu investigated the psychological conditions of 2042 medical staff and found that 25.5% had anxiety symptoms and 12.1% had depression symptoms [11]. Wang et al. reported that during the later periods of the pandemic, anxiety was still prevalent among medical staff, with 24.4% of the staff showing symptoms [12]. Pappa et al. conducted a meta-analysis of 13 studies (involving 33,062 medical staff), and the results indicated that 23.2% of medical staff exhibited anxiety and 22.8% had depression [13]. Marvaldi analysed 70 studies and found that around 30.0% of medical staff exhibited anxiety [14]. Preti et al. conducted a systemic evaluation of 44 studies, and the results indicated that 45% of medical staff had serious anxiety symptoms, 27.5~50.7% showed depression symptoms, and 34~36.1% had insomnia symptoms [15]. To study anxiety, Raoofi et al. conducted a meta-analysis of 46 articles that involved 61,551 hospital staff; the results indicated that 26.1% of medical staff suffered from anxiety; the proportions of healthcare technicians and medical students were 39% and 36%, respectively, indicating that a relatively higher proportion of hospital workers had anxiety [16]. The above literature review suggests that anxiety has been among the most broad and serious emotional symptoms of medical staff during the COVID-19 pandemic [17,18,19].

Moreover, medical staff who perform high-intensity work may experience reduced concentration, slower responses, a decline in the ability to understand and make judgements, and a decline in memory [20,21]. Research by Preti et al. [15]. indicates that during the COVID-19 pandemic, 34~36.1% of medical staff experienced insomnia. The long-term, high-intensity work also exhausted medical staff both physically and mentally. Certain staff experienced a high degree of job burnout, including a decreased sense of achievement, emotional exhaustion, and cynicism [22]. The above physical and mental symptoms also aggravate the anxiety and other emotional problems of front-line medical staff.

In recent years, network models have been proposed as an approach to conceptualise psychological construct and mental disorder. They provide an innovative method to mathematically analyse and visualise relations among complex variables. The traditional scale measurement of psychological constructs and mental disorders mostly uses the total-score model to judge the level or severity of symptoms. However, it is important to note that individuals with the same total score may have completely different combinations of symptoms. The total-score model ignores the characteristics of the heterogeneity of psychological constructs and mental disorders, which could be an important factor limiting the in-depth understanding of them. Different from the total-score model, a network structure consists of items and edges, which represent the variables and interconnections among variables, respectively [23]. The model treats psychological constructs and mental disorders as interactive systems; it is driven by data and considers that the interactions among the system’s composite parts actively participate in the creation of the construct, and that the interactions do not depend on the a priori hypothesis of the causal relationship between variables [24,25,26]. A network model can also provide indicators on centrality and predictability for each item to quantify its importance and controllability within the entire network [27]. The core variable in the psychological structure can be viewed as an important intervention goal and is believed to provide potential targets for relevant interventions. Predictability refers to the degree to which an item can be explained by adjacent items and is also an indicator of the extent of the network’s self-maintenance [21,27]. Community detection can be employed to identify communities of items, and items in the same community have a higher degree of interconnection than do items outside the community. The directed acyclic graph (DAG) is promising to provide reference for exploring the potential causality and core items (activation sources) from the perspective of probabilistic priority.

The network model provides a more fine-grained analysis for deepening the understanding of psychological constructs and mental disorders and also provides a reference for clinical intervention, which has important theoretical research and clinical application value. In view of this, an increasing number of studies are employing network models to analyse the network structure of mental constructs, for example, self-worth [28], resilience [29], personality, and so on [30]. Our project team has employed network analysis to study stigma [31], decision-making ability [32], comorbidity of anxiety and depression [33], and job burnout [34]. The research results mentioned above have been published. However, the samples in existing research that studies anxiety symptoms by applying network analysis have not included front-line medical staff who provided care for patients during the COVID-19 pandemic.

In summary, the aim of this study is to re-examine the anxiety symptoms of front-line medical staff who provided care for patients during the pandemic from a network perspective. Two different model-building methods—regularised partial correlation network (RPCN) and DAG—are used to perform a network analysis of anxiety from the symptomatology perspective, with a focus on the centrality indicators and probabilistic priority of each symptom. Additionally, the modes of connections among anxiety symptoms of front-line medical staff and the most influential symptoms are identified from the network perspective. It is expected that the findings will provide scientific evidence that facilitates a better in-depth understanding of the anxiety symptoms of front-line medical staff and the development of effective intervention measures.

## 2. Materials and Methods

### 2.1. Participants

The research participants were front-line medical staff in China who voluntarily participated in the study. The inclusion criteria were as follows: doctors and nurses in hospitals who participated in clinical work during the pandemic and signed an informed consent form regarding the contents of the study and voluntary participation. The exclusion criteria included incomplete questionnaires and overly long or short time for completing the questionnaire (response time beyond 3 standard deviations of the mean). This study was approved by the Clinical Trial Ethics Committee of the First Affiliated Hospital of Air Force Medical University and is registered at the Chinese Clinical Trial Registry, with the registration number CHiCTR1800019761.

From 18 February to 3 April 2020, the research team received responses from 14,578 medical staff. After excluding 186 invalid questionnaires, there were 14,392 valid questionnaires, representing 98.72% of all questionnaires received. Of the effective responses, 4509 were from front-line staff who participated in patient treatment. Previous research has indicated that a Generalized Anxiety Disorder 7-item (GAD-7) score of 8 or above is the recommended threshold for the clinical treatment of generalized anxiety; this indicator has a high degree of sensitivity and specificity [35,36]. As such, 1034 front-line medical staff who scored 8 or above on the GAD-7 questionnaire were selected as the participants for a further network structure analysis. Their general information is presented in Table 1.

### 2.2. Research Tools

#### 2.2.1. Questionnaire for Collecting General Information

Based on a literature review and expert evaluation, the research team developed the Questionnaire for Collecting General Information of Medical Staff during the COVID-19 Pandemic, which was designed to collect information regarding gender, marital status, education, job, title, and years of service (Table 1).

#### 2.2.2. GAD-7

The GAD-7 was developed by Spitzer et al. [37] with the quantification and assessment based on the Diagnostic and Statistical Manual of Mental Disorders, Fifth Edition (DSM-5) published by the American Psychiatric Association (APA) [38]. This measurement scale assesses the respondent’s state of anxiety by obtaining the frequency at which the respondent was bothered by anxiety symptoms during the previous 2 weeks. The measurement scheme comprises 7 symptoms: “Feeling nervous, anxious or on edge” (A1); “Not being able to stop or control worrying” (A2); “Worrying too much about different things” (A3); “Trouble relaxing” (A4); “Being so restless that it is hard to sit still” (A5); “Becoming easily annoyed or irritable” (A6); and “Feeling afraid, as if something awful might happen” (A7). Each question is scored using a four-point scale—0 “Not at all”; 1 “Several days”; 2 “More than half the days”; and 3 “Nearly every day”. The sum of the scores for each question is the total score, and a higher score indicates more serious anxiety symptoms [37]. The scale is simple in content, appropriate in number of questions, convenient for rapid evaluation, and has a good reliability, validity, sensitivity, and specificity, meeting the psychological measurement standard. In this study, the internal consistency of the questionnaire (Cronbach’α) was 0.74.

### 2.3. Research Process

Through the Mental Health Care Platform for Medical Staff, online sampling was used to administer the survey. Medical staff were asked to log in to the platform to complete the following: ① the informed consent form, which informed the subjects about the purpose, contents, and instructions and clarified the criteria for inclusion and exclusion; and ② the formal questionnaires, i.e., Questionnaire for Collecting General Information of Medical Staff during the COVID-19 Pandemic and the GAD-7. The participants’ responses to the questionnaires were anonymous so as to eliminate their concern and to prevent falsifying or concealing information. As the survey was administered online, if the participants completed the questionnaire, it was deemed that they had signed the informed consent form. The dates to complete the questionnaire were between 18 February and 3 April 2020. Each cell phone IP address could only be used to complete the questionnaire once. The survey settings only allowed the submission of the questionnaire when all questions were answered. All ethical rules and confidentiality requirements were strictly followed, and the findings were only be used for research purposes. Identifiable personal information was blurred when reporting the research findings. All experimental procedures were conducted in accordance with the principles outlined by the Declaration of Helsinki.

### 2.4. Statistical Analysis

Microsoft Excel was used to screen front-line medical staff who scored 8 or above on the GAD-7; the data were expressed as the mean ± standard deviation ($\overset{-}{x}$ ± s). R software (version 4.2.1) was employed to perform statistical and visual analyses.

The Kolmogorov–Smirnov test was employed to assess the distribution of the data for the 7 variables (symptoms) in the questionnaire, and the results indicated that the data had a non-normal distribution (significance level *p* < 0.05). As such, the Huge package was employed to perform non-paranormal transformation of the data [39,40], and the Kolmogorov–Smirnov test was conducted again to assess the distribution of the transformed data; the results indicated that the data still had a non-normal distribution. Therefore, nonparametric Spearman rho correlation matrices were used to input the network model, an approach supported by Epskam and Fried [39].

In addition, based on the methods employed in recent studies [41,42,43], we tested whether the correlation matrices were positive-definite matrices and applied the Hittner method (this approach was conducted through the goldbricker function in the networktools package [44]) to search possible overlapping variables [45]; the results indicated that the 7 variables of GAD-7 did not overlap.

### 2.5. RPCN

#### 2.5.1. Network Construction and Visualisation

Gaussian graphical models (GGMs) were employed to construct the network [46]. A GGM is an undirected network, and its edges represent the partial correlations between nodes [39]. The extended Bayesian information criterion (EBIC), combined with the graphical least absolute shrinkage and selection operator (gLASSO), was employed to select models, thus arriving at a more stable RPCN [47,48]. The gLASSO is a regularised algorithm; it can reduce very weak correlations to zero to control the number of false positive edges [39]. The EBIC employs a hyperparameter to control the model density [48]. Research indicates that when the hyperparameter is 0.5, it can well balance a network model’s explorations of the specificality and sensitivity of the true edges [49,50,51]. Therefore, the hyperparameter was set to 0.5, and the qgraph package was employed [52].

The Fruchterman–Reingold (FR) algorithm was used to visualise the network. The nodes of strong connections are located close to the centre of the network. Blue edges represent positive relationships between nodes, and red edges represent negative relations. The thickness of the edges denotes the strength of the connections between nodes [52,53]; this step was realised through the qgraph package [52].

#### 2.5.2. Centrality and Predictability

The expected influence (EI) that simultaneously considers edges with positive and negative correlations was used as the indicator to quantify the nodes’ relative importance [54,55]. A node’s EI is the sum of the weights of all edges connected to the node (i.e., the regularised partial correlation coefficients).

The mgm (Mixed Graphical Models) package was used to calculate each node’s predictability [27], which denotes to what extent a node’s variance can be predicted by the variances of the nodes connected to it; this measurement indicates whether the network is mainly determined by strong interactions among nodes or determined by other variables not in the network and can be used to describe the network’s controllability [56,57]. When a node has a high predictability, it can be controlled by other nodes in the network that are connected to the node; when a node has a low predictability, there is a need to find other variables outside the network or to directly interfere with the node itself to control it [56,57]. A precondition for determining node predictability is the assumption that all the edges that connect with that node point to the node; however, in cross-sectional data, the directions of edges are usually unknown. As such, node predictability was determined by the upper limit of the predictability of the nodes connected to it [27].

#### 2.5.3. Network Accuracy and Stability

The network accuracy and stability were estimated using the bootnet package [58,59]. First, a nonparametric bootstrap method (2000 bootstraps) was used to obtain the 95% confidence intervals (CI) for edge weights. The narrower the CI, the more accurate the estimate of edge weights and centrality indicators. Second, through a case-dropping bootstrap method, the correlation stability (CS) was calculated to estimate the stability of the nodes’ EI. Existing research indicates that the CS coefficient should be higher than 0.50 but not lower than 0.25 [58]. Lastly, through a bootstrap method (2000 times), a variance test was performed for the edge weights and EIs of nodes, respectively, to assess whether there existed significant differences between the 2 edge weights or between the EIs of 2 nodes (significance level α = 0.05).

#### 2.5.4. Community Test

The Spinglass algorithm of the igraph package was used to explore the community structure in the network [42,43,60]. This algorithm can identify whether the nodes form a single network structure or cluster into different communities. The rationale of the test is that the edges of the network should connect with nodes in the same community, and that nodes in different communities are not supposed to be connected [61]. To complement the results, we also used the walktrap algorithm, which is based on the principle that adjacent nodes tend to belong to the same community [34,61]. Other methods for identifying communities are referenced at https://psych-networks.com/r-tutorial-identify-communities-items-networks/ (accessed on 10 July 2023).

#### 2.5.5. Gender-Based Network Comparison

The NetworkComparisonTest package (version 2.2.2) was used to compare the networks of anxiety symptoms by gender (2000 iterations) [62]. Specifically, we conducted the global strength invariance test, edge invariance test, and centrality invariance test to examine whether there existed significant differences in terms of total edge weight, weight of the corresponding edges, and centrality of the corresponding nodes (the EI in this research), respectively, between anxiety symptom networks for males and females. As no a priori hypothesis was proposed for edge variance, multiple-comparison correction was not employed [62].

### 2.6. DAG

The Bayesian hill-climbing algorithm of the bnlearn package was employed to evaluate the DAG of anxiety symptoms [63,64]. Using the bootstrap function, this method can add or eliminate edges, or reverse the direction of edges, to ultimately optimise the Bayesian information criterion (BIC), thereby estimating the model’s structural characteristics. Based on the most current operational method for this bootstrap function, 50 random restarts were employed (i.e., using possible edges that connect different nodes), and 100 disturbances were applied to each restart (i.e., adding, eliminating, and reversing the direction of edges) [65,66,67]. When the iteration process for restart/disturbance progresses, the algorithm ultimately obtains the model with the optimal BIC.

To further ensure the stability of the final model, our operation steps were as follows. First, we bootstrapped 10,000 samples with replacements and evaluated a DAG for each sample. Second, we determined the frequency that the given edge appeared in the 10,000 samples. In this step, we employed the optimal cut-off method proposed by Scutari and Nagarajan to keep edges, so as to generate DAGs that had both a high sensitivity and specificity [68]. Only when the frequency at which the given edge appeared in the 10,000 DAGs exceeded the threshold determined through the optimal cut-off method would the edge be kept in the final DAG. Lastly, we determined the directions of the edges that were kept in the final DAG. If the directions of at least 51% of bootstrapped DAGs were from node A to B, then an arrow pointing from node A to B in the final DAG was used to represent the edge’s direction.

To facilitate the explanation, we visualised the DAG using approaches described in the most current literature [65,67]. In the first visualisation, the thickness of the arrow denotes a BIC change when the arrow is deleted from the DAG. The thicker the arrow, the greater the contribution of this edge to the model’s structure. In the second visualisation, the thickness of the arrow denotes the probability of the direction; that is, the proportion of the directions pointed to by this arrow in the bootstrap DAG. A thicker arrow indicates a larger proportion of the directions pointed to by that arrow in the DAG.

## 3. Results

### 3.1. Statistical Results for the Variables

The descriptive statistics for measuring the variable (anxiety symptoms) included the mean, standard deviation, maximum value, minimum value, EI, and predictability. Of the results, A1 “Feeling nervous, anxious or on edge” and A3 “Worrying too much about different things” were the most serious, and A5 “Being so restless that it is hard to sit still” was the least serious; see Table 2.

### 3.2. RPCN

#### 3.2.1. Network Structure

The RPCN of anxiety symptoms is shown in Figure 1.

There existed broad connections among anxiety symptoms (except that there were no connections between A1 “Feeling nervous, anxious or on edge” and A5 “Being so restless that it is hard to sit still”, A1 “Feeling nervous, anxious or on edge” and A7 “Feeling afraid as if something awful might happen”, and A2 “Not being able to stop or control worrying” and A6 “Becoming easily annoyed or irritable”, and there were a total of 18 edges among the nodes, accounting for 86% of the 21 maximum possible edges), and the connections were all positive correlations.

The strongest connections existed between A1 “Feeling nervous, anxious or on edge” and A2 “Not being able to stop or control worrying”, A6 “Becoming easily annoyed or irritable” and A7 “Feeling afraid as if something awful might happen”, A5 “Being so restless that it is hard to sit still” and A6 “Becoming easily annoyed or irritable”, A2 “Not being able to stop or control worrying” and A3 “Worrying too much about different things”, and A4 “Trouble relaxing” and A5 “Being so restless that it is hard to sit still”, and their regularized partial correlation coefficients were 0.40, 0.23, 0.23, 0.22, and 0.19, respectively. The connection strength among anxiety symptoms is provided in Table 3; the average strength was 0.10.

#### 3.2.2. Centrality and Predictability of Nodes

The EI of each anxiety symptom is shown in Figure 2. A2 “Not being able to stop or control worrying” had the largest EI (the raw score was 0.93), followed by A3 “Worrying too much about different things” (with a raw score of 0.67). A1 “Feeling nervous, anxious or on edge” had the smallest EI (with a raw score of 0.60). A2 “Not being able to stop or control worrying” had the highest predictability (0.37), indicating that 37% of the variance could be explained by the nodes connected to it. A4 “Trouble relaxing” had the lowest predictability (0.22), indicating that 22% of its variance could be explained by nodes connected to it. The predictability of anxiety symptoms is shown in Table 2; the average predictability was 0.25.

#### 3.2.3. Network Accuracy and Stability

Appendix A shows the results of using the bootstrap method to estimate the accuracy of edge weighs. The edge weights’ CI derived through the bootstrap method was relatively narrow. Considering there were a large number of participants in this study (1034 participants) and few variables (seven variables), the CI indicated that the edge weights were accurate.

The difference test results of edge weights are shown in Appendix A. The difference test was used to assess whether there was a statistically significant difference between two edge weights. The results indicated that the strength of the connection of the five edges with the strongest connections was significantly greater than the strength of the rest of the edges at 59–100% (there was a significant difference between edge A4 “Trouble relaxing”–A5 “Being so restless that it is hard to sit still” and 10 of the rest of the 17 edges and between edge A1 “Feeling nervous, anxious or on edge”–A2 “Not being able to stop or control worrying” and the rest of the 17 edges).

The CS coefficient of anxiety symptoms’ EI is shown in Appendix A. The coefficient was 0.60, indicating that the anxiety symptoms’ EI in this research was sufficiently stable.

The results of the difference test of the nodes’ EI are shown in Appendix A; these findings helped evaluate whether there existed a significant difference between two nodes’ EI. The EI of A2 “Not being able to stop or control worrying” was significantly greater than the EI of other symptoms, and there was no significant difference among other symptoms’ EI.

#### 3.2.4. Community Test

The Spinglass algorithm and walktrap algorithm identified the same two communities. One community consisted of A1 “Feeling nervous, anxious or on edge”, A2 “Not being able to stop or control worrying”, and A3 “Worrying too much about different things”, and the other consisted of A4 “Trouble relaxing”, A5 “Being so restless that it is hard to sit still”, A6 “Becoming easily annoyed or irritable”, and A7 “Feeling afraid as if something awful might happen”.

#### 3.2.5. Gender-Based Network Comparison

The results indicated that the total weight of edges (overall strength) in the females’ anxiety symptom network was greater than that in the males’ anxiety symptom network, but the difference was not statistically significant (overall strength [S] = 0.16, male = 2.27, female = 2.43, *p* = 0.43). The edge invariance test and centrality invariance test indicated that there was no significant difference in the weight of the corresponding edges or in the centrality of the corresponding nodes between the anxiety symptom networks for males and females.

### 3.3. DAG

In Figure 3, the thickness of the arrow indicates that the BIC changes when the arrow is deleted from the DAG. The thicker the arrow, the greater its contribution to the model. The most important arrows in the DAG were (1) from A2 “Not being able to stop or control worrying” to A1 “Feeling nervous, anxious or on edge” (BIC change: −152.35); (2) from A2 “Not being able to stop or control worrying” to A7 “Feeling afraid as if something awful might happen” (BIC change: −41.76); and (3) from A2 “Not being able to stop or control worrying” to A4 “Trouble relaxing” (BIC change: −41.62). The BIC changes when each arrow in Figure 3 is deleted are shown in Table 4.

In Figure 4, the thickness of the arrow denotes the probability of the direction, that is, the proportion of the directions pointed to by the arrow in the bootstrapped DAG. The thicker the arrow, the larger the proportion of the directions pointed to in the bootstrapped DAG. The thickest arrows were (1) A7 to A6 (probability: 0.65; that is, of the 10,000 bootstrapped DAGs, 6500 DAGs were in this direction); (2) from A2 to A3 (probability: 0.65); and (3) from A5 to A6 (probability: 0.65). Each arrow’s probability of direction is shown in Table 4.

## 4. Discussion

Two different model-building methods—the RPCN and DAG—were employed to construct the anxiety symptom network for front-line medical staff during the COVID-19 pandemic, and each symptom’s centrality indicators and probabilistic priority (potential causal relation) were examined. From the results pertaining to the connection modes of the RPCN and DAG, centrality analysis, probabilistic priority, predictability, community test, and gender-based network comparison, we could determine which symptoms were closely connected and likely to form a “subsystem”, identify the core symptoms and upstream symptoms in the anxiety symptom network of front-line medical staff at the beginning of the pandemic, and determine whether the anxiety symptom network had gender-based differences. These results may be of help in the prevention of and interventions for anxiety symptoms in front-line medical staff during the COVID-19 pandemic.

### 4.1. RPCN

#### 4.1.1. Network Structure

The strongest connections existed between A1 “Feeling nervous, anxious or on edge” and A2 “Not being able to stop or control worrying”, A6 “Becoming easily annoyed or irritable” and A7 “Feeling afraid as if something awful might happen”, A5 “Being so restless that it is hard to sit still” and A6 “Becoming easily annoyed or irritable”, A2 “Not being able to stop or control worrying” and A3 “Worrying too much about different things”, and A4 “Trouble relaxing” and A5 “Being so restless that it is hard to sit still”. In fact, there existed strong connections between A2 “Not being able to stop or control worrying” and A1 “Feeling nervous, anxious or on edge” and between A2 “Not being able to stop or control worrying” and A3 “Worrying too much about different things” in an anxiety symptom network for the Chinese public during the COVID-19 outbreak [69], for Chinese university students during the later period of the pandemic [70], for Norwegian adults during pandemic lockdowns [71], and for the British public during the pandemic [72]. This might be an indication that the connections between A2 “Not being able to stop or control worrying” and A1 “Feeling nervous, anxious or on edge” and between A2 “Not being able to stop or control worrying” and A3 “Worrying too much about different things” are consistent and stable across different pandemic periods, different groups of people, and different countries. As such, these may be connections of symptom worthy of greater attention by researchers and experts. In addition, a 2016 study on the anxiety symptom network of mental disorder patients also found that there existed close connections between A2 “Not being able to stop or control worrying” and A1 “Feeling nervous, anxious or on edge” and between A2 “Not being able to stop or control worrying” and A3 “Worrying too much about different things” [47]. There also existed strong connections between A4 “Trouble relaxing” and A5 “Being so restless that it is hard to sit still” and between A5 “Being so restless that it is hard to sit still” and A6 “Becoming easily annoyed or irritable”; this finding is consistent with the anxiety symptom network for Belgian people during the pandemic [42], for the Chinese public during the pandemic [69], and for Norwegian adults [71].

A2 “Not being able to stop or control worrying” focuses on the unconscious and uncontrollable perception of the event being worried about; A1 “Feeling nervous, anxious or on edge” focuses on the description of emotional nervousness and anxiety caused by cognition and other factors; and A3 “Worrying too much about different things” focuses on reflecting the broadness of worries, covers broad content, and does not address a specific aspect. The outbreak of COVID-19 was sudden, and it spread rapidly; it caused a large number of infections and deaths in a short period of time. At the beginning of the pandemic, there was no specific medicine, vaccines, effective prevention measures that could treat the disease; even personal protective equipment (PPE) was in short supply [73]. Furthermore, there was never-ending reporting on the pandemic; the daily number of new patients and deaths increased rapidly; and news about lockdowns were omnipresent. With long-term exposure to this dangerous medical care environment and constant bombardment of pandemic-related information, medical staff unconsciously and uncontrollably develop anxiety [20,22]. Two aspects may drive this anxiety. First, front-line medical staff may worry about becoming infected when treating patients, their family’s health and life, and whether the pandemic will worsen, i.e., broad, nonspecific worries about pandemic-related events. Second, they would inevitably be nervous or have feelings of urgency with regard to access to PPE or interacting with their family. Excessively worrying about various matters and feelings of nervousness, anxiousness, or urgency can contribute to losing control over the concerns. As the connections among symptoms in the RPCN are directionless [39], the interconnections between A2 “Not being able to stop or control worrying” and A1 “Feeling nervous, anxious or on edge” and between A2 “Not being able to stop or control worrying” and A3 “Worrying too much about different things” require further investigation.

#### 4.1.2. Centrality and Predictability of Item

A2 “Not being able to stop or control worrying” had the strongest EI, indicating that this symptom had the broadest and closest connection with other anxiety symptoms in the network; in other words, its activation was more likely to spread throughout the entire anxiety symptom network. As such, interventions aimed at A2 “Not being able to stop or control worrying” would more effectively reduce the overall level of anxiety symptoms than would interventions aimed at other symptoms. This provides an important target for the prevention of and interventions for the anxiety symptoms in front-line medical staff during the pandemic. Existing research on anxiety symptom networks has also indicated that A2 “Not being able to stop or control worrying” is the core node of anxiety network structures and comorbidity network structure of anxiety and depression [72,74,75].

The results of the predictability analysis suggested that A2 “Not being able to stop or control worrying” had the highest predictability and that 37% of its variance could be explained by items connected to it, an indication that this symptom was less influenced by other items in the network that are connected to it. The implication of this finding is that while A2 “Not being able to stop or control worrying” can be modified through interfering with the nodes that are closely connected to it, more emphasis should be placed on measures that target A2 “Not being able to stop or control worrying” itself or other variables outside of the network (e.g., social or biological factors) to modify A2 “Not being able to stop or control worrying”. Notably, this was a cross-sectional study, and the directions of edges were unknown. As such, a node’s predictability was determined by the upper limit of the predictability of the nodes connected to it [54]. The average predictability of symptoms in the anxiety network was 0.26, lower than the average predictability of the anxiety symptom network for the Belgian people during the pandemic lockdowns [42]. This may have been caused by the sample selection method for this research, whereby only the responses of participants with a GAD-7 score of eight or above were included—a higher score indicates that more symptom combinations are needed to arrive at the same total score and that the degree of connection among symptoms is lower.

#### 4.1.3. Community Detection

The results of the community detection indicated that the seven symptoms formed two communities (clusters of symptoms). A1 “Feeling nervous, anxious or on edge”, A2 “Not being able to stop or control worrying”, and A3 “Worrying too much about different things” were community one, mainly focused on the cognitive symptoms of anxiety. A4 “Trouble relaxing”, A5 “Being so restless that it is hard to sit still”, A6 “Becoming easily annoyed or irritable”, and A7 “Feeling afraid as if something awful might happen” formed another community, focusing on the physical and emotional symptoms, for example, “it is difficult to relax” and “I am too upset to sit still”. Our study validated the reliability of the symptomatic diagnostic criteria for GAD in the DSM-5 [38] from a network perspective as follows: A3 “Worrying too much about different things” essentially corresponded with core symptom A in the diagnostic criteria, i.e., “Excessive anxiety and worry (apprehensive expectation), occurring more days than not for at least 6 months, about several events or activities (such as work or school performance)”; A2 “Not being able to stop or control worrying” corresponded with core symptom B, i.e., “The individual finds it difficult to control the worry”; and A4 “Trouble relaxing”, A5 “Being so restless that it is hard to sit still”, A6 “Becoming easily annoyed or irritable”, and A7 “Feeling afraid as if something awful might happen” essentially corresponded with accompanying symptoms C, i.e., “Restlessness or feeling keyed up or on edge, being easily fatigued, difficulty concentrating or mind going blank, irritability, muscle tension, and sleep disturbance”. Core symptoms A and B are necessary, while accompanying symptoms C need to meet at least three symptoms (only one for children) [68]. This may be because, in diagnosing GAD, core symptoms A and B must be met simultaneously, while accompanying symptoms may occur in different combinations. Therefore, there is a greater possibility of the co-occurrence of core symptoms A and B among front-line medical staff who scored eight or above on the GAD-7, resulting in a higher degree of correlation; as a result, the symptoms clustered into a community. The close connections among A1 “Feeling nervous, anxious or on edge”, A2 “Not being able to stop or control worrying”, and A3 “Worrying too much about different things” can also be found in the RPCN. In the community detection of the anxiety symptom network for Belgian people during the pandemic lockdowns, the seven symptoms in the GAD-7 belonged to the same community; namely, the seven symptoms existed as a single network structure [42]. This is different from the findings of this study, a difference that may have been caused by the uniqueness of the sample used in this research (i.e., subjects all scored eight or above on the GAD-7) and is worthy of further investigation through future research.

#### 4.1.4. Gender-Based Network Comparison

Gender difference is an important factor in the study of medical staff’s mental health and has increasingly become a social issue and an important qualification for some jobs. It is commonly believed that females are more gentle, attentive, approachable, and perform better in clinical and nursing work. Generally speaking, females account for a large portion of nurses; the gender composition of medical staff in this study supports that fact [76]. The results of the anxiety symptom network comparison indicates that although gender-based difference was not significant, the overall connection in females’ anxiety symptom network was closer. Previous research also indicated that females were more likely to suffer from anxiety [77,78]. In addition, no significant gender-based differences in terms of the weight of the corresponding edges and centrality of the corresponding items were found. The above findings may indicate that when faced with major event stressors, such as the COVID-19 pandemic, the connection modes for symptoms may be the same for both genders.

### 4.2. DAG

The DAG indicated that A2 “Not being able to stop or control worrying” was at the top of the GAD symptoms; in other words, from a probability and statistics perspective, A2 “Not being able to stop or control worrying” was the activation source of the GAD symptom and had the highest probabilistic priority (i.e., potential causal relationship). The implication of this finding is that under normal conditions, front-line medical staff are unlikely to demonstrate other anxiety symptoms; however, when A2 “Not being able to stop or control worrying” occurs, other symptoms are likely. This finding is consistent with the results of the centrality analysis in the RPCN and provides supplemental evidence that A2 “Not being able to stop or control worrying” is the core symptom in the GAD network. Although the assumptions and constraining conditions of the two model-building methods, i.e., RPCN and DAG, are different, the results are consistent; that is, the results highlight the centrality of A2 “Not being able to stop or control worrying” in the GAD network. In the DAG, A2 “Not being able to stop or control worrying” directly points to A1 “Feeling nervous, anxious or on edge”, A3 “Worrying too much about different things”, A4 “Trouble relaxing”, A5 “Being so restless that it is hard to sit still”, and A7 “Feeling afraid as if something awful might happen”; that is, the existence of A2 “Not being able to stop or control worrying” means there is a greater possibility that the above symptoms exist or that most symptom conditional probabilities of the GAD depend on the presence of A2 “Not being able to stop or control worrying”. This is also similar to the symptom connection modes in the RPCN. Although there was no direct probability dependence between A2 “Not being able to stop or control worrying” and A6 “Becoming easily annoyed or irritable”, indirect probability dependence can be created through other paths, for example, from A2 “Not being able to stop or control worrying” to A7 “Feeling afraid as if something awful might happen” and to A6 “Becoming easily annoyed or irritable”, and from A2 “Not being able to stop or control worrying” to A1 “Feeling nervous, anxious or on edge” to A3 “Worrying too much about different things” and to A6 “Becoming easily annoyed or irritable”.

In fact, in the DAG in which the edge thickness represents the probability of the direction, the fact that the thickness of the edge denotes the probability of the direction pointed to by the edge does not mean there does not exist a probability that the edge points to another direction [65]. In this paper’s DAG, most edges were thin. For example, in only 55% of the bootstrap networks, the edge from A2 “Not being able to stop or control worrying” to A1 “Feeling nervous, anxious or on edge” pointed to the direction described by it; and only in 51% of the bootstrap networks did the edge from A2 “Not being able to stop or control worrying” to A1 “Feeling nervous, anxious or on edge” point to the direction described by it. Therefore, the predicted directions among these variables may be two-way, and this should be noted in explaining the connection modes among symptoms.

The DAG can provide unique ideas for clinical interventions—upstream symptoms have a higher probabilistic priority and should be the major targets of interventions; prioritising interventions for upstream symptoms will generate the broadest effects. In other words, prioritising interventions for A2 “Not being able to stop or control worrying” may create the most beneficial effects with regard to alleviating the anxiety symptom group of front-line medical staff who scored eight or above on the GAD-7. This finding is consistent with the result of the RPCN and to a certain extent reinforces our confidence in prioritising interventions for A2 “Not being able to stop or control worrying” to ease GAD symptoms.

This research has certain limitations. First, the participants who scored eight or above on the GAD-7 are not equal to clinical patients [35,36]. The symptoms and timeframe of anxiety measured by the GAD-7 does not completely coincide with the GAD diagnoses in the DSM-5; therefore, the research findings may only apply to a certain group of people at a particular time. Second, this was a cross-sectional study; even if the DAG was employed to investigate the directions of connections among variables, the results do not represent priorities in the timeline and should not be explained as causal effects [79,80]. In future research, ecological momentary assessment and time series data may be used to explore the causal relationship among items over time [81]. Third, the DAG assumes that the relationship among variables is acyclic, but a feedback loop among symptoms is very likely to play a major role in mental illness symptomology [82,83]. Therefore, the DAG cannot be used to investigate the feedback loop among symptoms. Fourth, the network in this study estimated the influences among participants at the group level, whereas symptom networks with different GAD total scores (e.g., 8, 9, 10, etc.) may have different structures; at the individual level, characteristics such as centrality and network structure may differ. In addition, nonprobabilistic sampling methods may make individuals with specific interests or experiences more inclined to participate, resulting in a certain degree of self-selection bias, which limits the generalizability of research results. Fifth, the network structure is limited by items in the network; therefore, only variables entered in the model are analysed, and thus, other anxiety symptoms or variables closely related to anxiety symptoms may be omitted. In addition, there may exist certain differences among the survey results obtained using different anxiety questionnaires; the State-Trait Anxiety Inventory [84], GAD-7 [42,85], and Beck Anxiety Inventory [86] may have different network structure characteristics, which should be investigated in future research.

## 5. Conclusions

In the current study, a network analysis was used to deepen the understanding of GAD among front-line medical staff during the COVID-19 pandemic and provide a reference for intervention targets. The results showed that there was a very wide correlation between the anxiety symptoms. A2 “Not being able to stop or control worrying” may be the most critical core symptom, which plays an important role in the occurrence and development of the overall anxiety situation and other anxiety symptoms. At the same time, A2 “Not being able to stop or control worrying” is also considered to be an important intervention target, that is, an intervention on A2 “Not being able to stop or control worrying” to minimize the anxiety symptom cluster. In addition, the symptoms of GAD also showed “subsystems” and gender characteristics.

## Figures and Tables

**Figure 1 brainsci-13-01155-f001:**
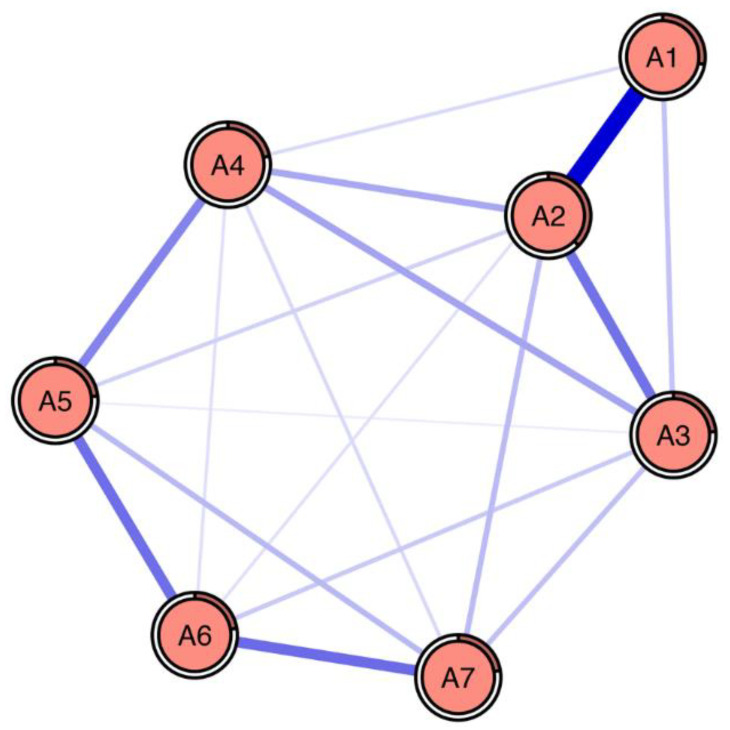
Regularized partial correlation network (RPCN) of GAD-7 anxiety symptoms. Note: The blue lines represent the positive relationship. The thicker the edge, the greater the correlation between the two variables; the thinner the edge, the smaller the correlation between the two variables. The ring around a node describes the degree of its predictability. A1 “Feeling nervous, anxious or on edge”; A2 “Not being able to stop or control worrying”; A3 “Worrying too much about different things”; A4 “Trouble relaxing”; A5 “Being so restless that it is hard to sit still”; A6 “Becoming easily annoyed or irritable”; A7 “Feeling afraid as if something awful might happen”.

**Figure 2 brainsci-13-01155-f002:**
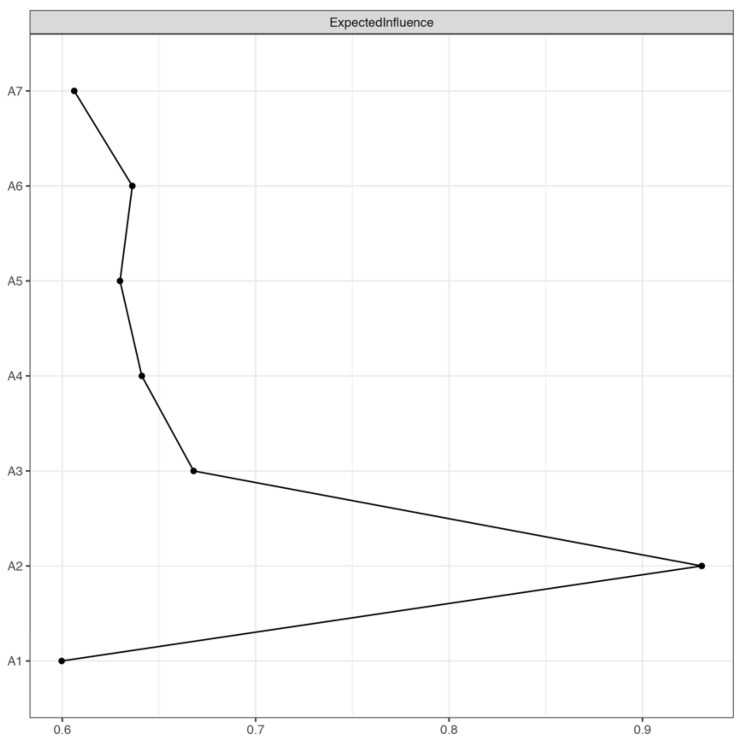
Expected influence of GAD-7 anxiety symptoms (raw score). Note: A1 “Feeling nervous, anxious or on edge”; A2 “Not being able to stop or control worrying”; A3 “Worrying too much about different things”; A4 “Trouble relaxing”; A5 “Being so restless that it is hard to sit still”; A6 “Becoming easily annoyed or irritable”; A7 “Feeling afraid as if something awful might happen”.

**Figure 3 brainsci-13-01155-f003:**
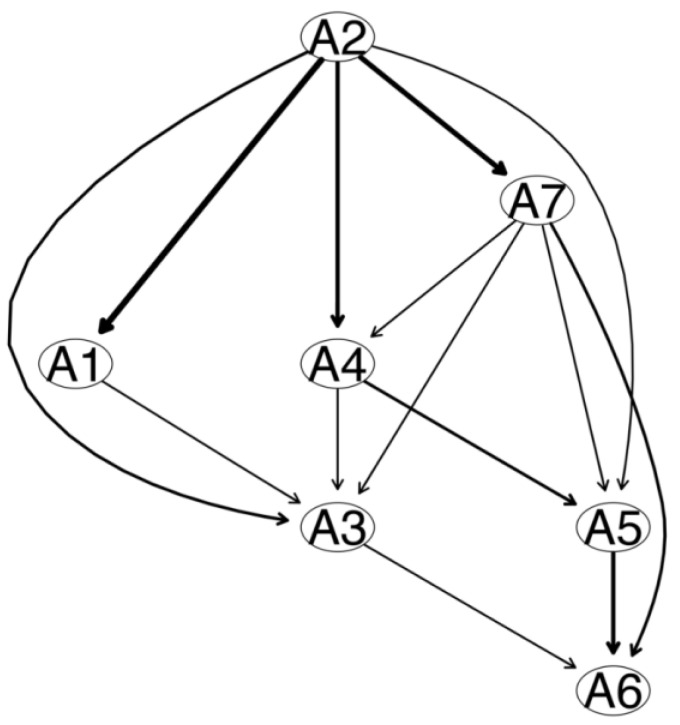
Directed acyclic graph (DAG) of GAD-7 anxiety symptoms (the thickness represents the importance of the model). Note: A1 “Feeling nervous, anxious or on edge”; A2 “Not being able to stop or control worrying”; A3 “Worrying too much about different things”; A4 “Trouble relaxing”; A5 “Being so restless that it is hard to sit still”; A6 “Becoming easily annoyed or irritable”; A7 “Feeling afraid as if something awful might happen”.

**Figure 4 brainsci-13-01155-f004:**
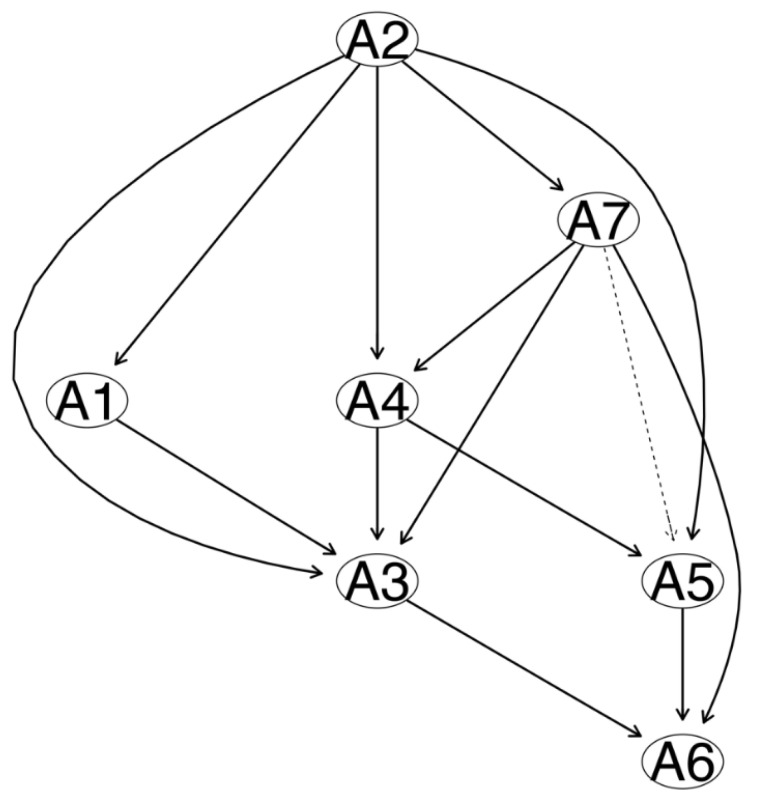
Directed acyclic graph (DAG) of GAD-7 anxiety symptoms (the thickness represents the direction probability). Note: The dotted line represents the direction probability less than 51%, that is, the direction probability from A7 to A5 is 0.50, see Table 4. Note: A1 “Feeling nervous, anxious or on edge”; A2 “Not being able to stop or control worrying”; A3 “Worrying too much about different things”; A4 “Trouble relaxing”; A5 “Being so restless that it is hard to sit still”; A6 “Becoming easily annoyed or irritable”; A7 “Feeling afraid as if something awful might happen”.

**Table 1 brainsci-13-01155-t001:** General information of first-line medical staff with GAD ≥ 8 under the COVID-19 pandemic.

Item	Number (*n*)	Percentage (%)
Gender		
Male	238	23.0
Female	796	77.0
Marital status		
Married	825	79.8
Unmarried	189	18.3
Other (Widowhood, divorce, etc.)	20	1.9
Education		
Below bachelor’s degree	340	32.9
Bachelor’s degree or above	694	67.1
Job		
Doctor	302	29.2
Nurse	732	70.8
Title		
Non	91	8.8
Primary	444	42.9
Intermediate	341	33.0
Senior	158	15.3
Years of service		
≤3 years	181	17.5
4–10 years	346	33.5
≥10 years	507	49.0

Note: Personnel without professional title were interns, volunteers and some people who did not fill in the professional title.

**Table 2 brainsci-13-01155-t002:** Descriptive statistics of GAD-7 anxiety symptoms.

	$\overset{-}{x}$ ± s	Min	Max	Expected Influence (EI)	Predictability (Pre)
A1	1.86 ± 0.81	0	3	0.60	0.28
A2	1.69 ± 0.78	0	3	0.93	0.37
A3	1.85 ± 0.75	0	3	0.67	0.24
A4	1.80 ± 0.78	0	3	0.64	0.22
A5	1.28 ± 0.78	0	3	0.63	0.23
A6	1.76 ± 0.82	0	3	0.64	0.22
A7	1.33 ± 0.78	0	3	0.61	0.22

Note: A1 “Feeling nervous, anxious or on edge”; A2 “Not being able to stop or control worrying”; A3 “Worrying too much about different things”; A4 “Trouble relaxing”; A5 “Being so restless that it is hard to sit still”; A6 “Becoming easily annoyed or irritable”; A7 “Feeling afraid as if something awful might happen”.

**Table 3 brainsci-13-01155-t003:** Regularized partial correlation between GAD-7 anxiety symptoms.

	A1	A2	A3	A4	A5	A6	A7
A1	0						
A2	0.40	0					
A3	0.10	0.22	0				
A4	0.06	0.13	0.14	0			
A5	0	0.07	0.03	0.19	0		
A6	0.05	0	0.08	0.05	0.23	0	
A7	0	0.11	0.10	0.06	0.11	0.23	0

Note: A1 “Feeling nervous, anxious or on edge”; A2 “Not being able to stop or control worrying”; A3 “Worrying too much about different things”; A4 “Trouble relaxing”; A5 “Being so restless that it is hard to sit still”; A6 “Becoming easily annoyed or irritable”; A7 “Feeling afraid as if something awful might happen”.

**Table 4 brainsci-13-01155-t004:** Weight value in the directed acyclic graph (DAG) represented by each arrow.

Arrow		Thickness Value of Arrow	
From	To	BIC	Direction Probability
A1	A3	−3.24	0.57
A2	A1	−152.35	0.55
A2	A3	−25.17	0.65
A2	A4	−41.62	0.62
A2	A5	−3.34	0.64
A2	A7	−41.76	0.64
A3	A6	−3.22	0.63
A4	A3	−10.62	0.51
A4	A5	−27.35	0.55
A5	A6	−35.22	0.65
A7	A3	−7.78	0.52
A7	A4	−13.05	0.53
A7	A5	−22.93	0.50
A7	A6	−33.83	0.65

Note: A1 “Feeling nervous, anxious or on edge”; A2 “Not being able to stop or control worrying”; A3 “Worrying too much about different things”; A4 “Trouble relaxing”; A5 “Being so restless that it is hard to sit still”; A6 “Becoming easily annoyed or irritable”; A7 “Feeling afraid as if something awful might happen”.

## Data Availability

The raw data supporting the conclusions of this article will be made available from the corresponding author on reasonable request.

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
