# Peer review of "Network Analysis of Anxiety Symptoms in Front-Line Medical Staff during the COVID-19 Pandemic"

_brainsci, 2023, doi:10.3390/brainsci13081155_

Round 1

Reviewer 1 Report

Comments and Suggestions for Authors

The manuscript provides a comprehensive exploration of the experiences of anxiety symptoms among front-line medical staff during the challenging times of the COVID-19 pandemic, using an innovative network analysis approach. While the study offers valuable insights, there are some areas that require clarification and improvement.

Introduction:

The introduction offers a comprehensive overview of the impact of the COVID-19 pandemic on the well-being of medical staff, shedding light on the cognitive, behavioral, and emotional challenges they face. However, it would be helpful to clearly state the research objective or question at the beginning, ensuring that readers can readily understand the purpose of the study. Additionally, a concise explanation of the significance and advantages of using network models would enhance readers' understanding of the chosen approach (see also my comments in the discussion section). It would also be beneficial to provide a rationale for including front-line medical staff in the study, addressing the existing research gap, and highlighting the importance of focusing on this specific population.

Methods:

The authors opted for a nonprobabilistic sampling method, allowing medical staff to participate voluntarily through the Mental Health Care Platform. While this approach provides valuable insights, it is important to acknowledge the potential for self-selection bias, as individuals with specific interests or experiences may be more likely to participate. To enhance the validity and generalizability of the study, reporting data on the acceptance rate and discussing the limitations of the sampling method would be prudent. Furthermore, clarifying whether the analyses involved only the screened participants, or the entire collected sample is necessary, as the excluded participants (or part of them) could potentially serve as a valuable control group for comparison. Exploring the network structure differences between included and excluded participants could provide meaningful insights.

The authors employed different numbers of resampling (e.g., 2000 and 10000) for bootstrap analysis without providing a clear rationale. Why always a different number of replications used? Providing an explanation for this choice or acknowledging that the selection was arbitrary would enhance the transparency of the methodology. Including a rationale to support the chosen resampling numbers would further strengthen the study's methodological rigor.

Regarding the Spinglass algorithm, it is advisable to address its inherent variability by iterating the procedure with different seeds. Additionally, comparing the Spinglass communities with those generated by other algorithms would offer a more comprehensive understanding of the network structure. Referring to additional resources, such as the provided link on identifying communities in networks, would greatly assist readers in comprehending the methodology.

https://psych-networks.com/r-tutorial-identify-communities-items-networks/

Discussion:

The manuscript appropriately acknowledges the limitations of establishing causal relationships in cross-sectional studies. While the DAG analysis holds promise for detecting directionality, it is crucial to emphasize that causality cannot be established solely based on cross-sectional data. In the introduction, it is important to clearly communicate the heuristic value of the DAG in this context, managing reader expectations and ensuring a nuanced interpretation.

General Comments:

Throughout the manuscript, the use of symptom codes (A1-A7) presents challenges in following the results and discussion. Replicating Table 2 as a caption in the figures and replacing the codes with item descriptions would significantly improve readability and comprehension. Also, using brief labels in the text can improve readability. For example, at first glance, I found the abstract totally uninformative because I had no idea of the contents of A1-A7.

The network analyses were based on at least three different r packages. Since the data did not meet the requirements for parametric analyses, I wonder if the authors set Spearman correlations or a nonparanormal transformation in the options for each package. It would be helpful if r codes were reported among the supplementary materials for greater transparency and replicability of the results.

Conclusion

Overall, the manuscript contributes to the literature on anxiety symptoms among front-line medical staff during the COVID-19 pandemic. By addressing the mentioned areas and refining the clarity of the presentation, the manuscript will become more accessible and impactful.

Author Response

Thank you very much for the reviewer's recognition and guidance on our work. Your feedback is very valuable and has given us great inspiration. We have revised the manuscript according to your feedback. Please feel free to contact us if there are any modifications needed in the manuscript.

We hope that our manuscript can meet the publishing standards of 'Brain Sciences'.

Once again, thank you very much for your comments and suggestions.

Reviewer 2 Report

Comments and Suggestions for Authors

There are several minimal errors that should be checked.

In page 5, line 202

Please, write complete the significance of acronym RPCN

In page 5, line 207

Please, write complete the significance of acronym gLASSO

In page 5, line 224

Please, write complete the significance of acronym mgm

In page 8, line 305, in Figure 1

Note: The blue lines represent the positive relationship; the red lines represent the negative relationship.

But, in this figure, I did not see red lines, ¿could you check what is the real color?.

Author Response

(The authors gave the same response as above.)

Round 2

Reviewer 1 Report

Comments and Suggestions for Authors

Dear Editor, the authors have been responsive to my comments and have diligently addressed all the issues raised during the review process.

I am satisfied with the revision and believe the manuscript is suitable for publication.

I commend the authors for their efforts in enhancing the quality of the manuscript.